# Separating Curing and Temperature Effects on the Temperature Coefficient of Resistance for a Single-Walled Carbon Nanotube Nanocomposite

**DOI:** 10.3390/polym15020433

**Published:** 2023-01-13

**Authors:** Milad Jafarypouria, Biltu Mahato, Sergey G. Abaimov

**Affiliations:** Center for Petroleum Science and Engineering, Skolkovo Institute of Science and Technology, Bolshoy Boulevard 30 bld. 1, Moscow 121205, Russia

**Keywords:** temperature coefficient of resistance, CNT/epoxy nanocomposite, post-curing

## Abstract

The temperature coefficient of resistance (*TCR*) determines the electrical performance of materials in electronics. For a carbon nanotube (CNT) nanocomposite, change of resistivity with temperature depends on changes in CNT intrinsic conductivity, tunnelling thresholds and distances, matrix’ coefficient of thermal expansion, and other factors. In our study, we add one more influencing factor–the degree of cure. Complexities of the curing process cause difficulties to predict, or even measure, the curing state of the polymer matrix while uncertainty in the degree of cure influences *TCR* measurements leading to biased values. Here we study the influence of the cure state on the *TCR* of a single-walled CNT/epoxy polymer nanocomposite. For the given degree of cure, *TCR* measurements are conducted in the temperature range 25–100 °C, followed by the next 24 h of post-curing and a new cycle of measurements, 8 cycles in total. We find that contrary to industry practice to expect a high degree of cure after 3 h at 130 °C, the curing process is far from reaching the steady state of the material and continues at least for the next 72 h at 120 °C, as we observe by changes in the material electrical resistivity. If *TCR* measurements are conducted in this period, we find them significantly influenced by the post-curing process continuing in parallel, leading in particular to non-monotonic temperature dependence and the appearance of negative values. The unbiased *TCR* values we observe only when the material reaches the steady state are no longer influenced by the heat input. The dependence becomes steady, monotonically increasing from near zero value at room temperature to 0.001 1/°C at 100 °C.

## 1. Introduction

Complimentary to good mechanical performance, epoxy resins suffer from poor electrical conductivities [1,2]. Carbon nanotubes (CNTs) enhance polymer properties as a nanofiller and have been promising candidates to increase the electrical conductivities of epoxy-based nanocomposites with high precision and piezoresistive sensitivity [3,4,5,6,7]. Especially in the field of high-precision sensors [8,9,10], the influence of environment temperature [11,12] cannot be ignored for multiple polymer functionalities. Considering electrical properties, this phenomenon is commonly related to the temperature coefficient of resistance (*TCR*).

*TCR* determines the performance of electrical materials in electronics. *TCR* accurate values are required for the design of electronic components, especially in high-precision devices. *TCR* is defined as
(1)$$TCR=\frac{1}{R}\frac{dR}{dT}$$
where *R* is the electrical resistance (or resistivity) of the material and *dR* is its change with the change in temperature *dT*.

*TCR* of a CNT/polymer nanocomposite depends extremely on the morphology of CNT networks and properties of CNTs and polymer [13]. The CNT intrinsic conductance and tunnelling (or contact) conductance between CNTs are the two basic types of conductance determining the electrical response of a CNT/ polymer nanocomposite [14,15,16,17,18]. Both are complementary one to another [9] when the leading mechanism of resistance is determined by many factors, first of all, chirality. Besides, the electrical behaviour of CNT networks is significantly influenced by CNT shapes, CNT wall count, functional surface groups, bundling, agglomeration, and CNT deformation at contacts–phenomena that are difficult to be described quantitatively. For example, experimental observations show that the interactions of tube-tube or tube-matrix could lead to structural distortions of CNTs [19,20,21]. It could substantially affect the local electrical structure and act as a strong scatter to reduce the intrinsic conductance. CNT shapes, described by curvature and torsion of their centerlines, also significantly influence the nanocomposite resistance [18]. Often studies assume a ballistic limit of infinite CNT intrinsic conductance, but detailed analysis [9] demonstrates that more experimental evidence is required, especially since experiments are often biased by CNT bundling (van der Waals attraction causing single-walled CNTs to form bundles) and CNT agglomeration (entanglement). Moreover, the investigations of humidity effects [22,23,24,25] show that the resistivity of CNT networks exponentially depends on the development of water absorption. All mentioned factors influence the *TCR* of a studied material leading to complex analyses in support of experimental observations.

As electrical materials, CNT nanocomposites find wide applications nowadays with several studies addressing the *TCR*s of these materials. Approaches to effectively tailor the *TCR* of CNT/polymer nanocomposites, including zero and/or enhanced *TCR*, have been developed. Skákalová et al. [26] measured and compared the electronic transport properties of individual multi-walled carbon nanotubes (MWCNTs) and individual single-walled carbon nanotubes (SWCNTs), as well as their networks of varying thickness, not infused with polymer. They found mostly negative *TCR*s at low temperatures below room temperature but argued about the presence of a peak in conductance-temperature dependence at high temperatures. Negative *TCR*s were as well reported for non-infused aligned CNT forests by Lee et al. [27]. Gong et al. [13,28] investigated the influence of temperature on the MWCNT/epoxy nanocomposite *TCR*s as critical for high-accuracy sensors. Observing complex temperature behaviour, the authors argued it resulted from the competition of the tunnelling and thermal expansion of the polymer matrix. Varying the loading of CNTs, the authors tailored the nanocomposite to the near-zero *TCR* in a wide temperature range.

For MWCNT/vinyl ester nanocomposites, studied at different fractions of nanofiller, Lasater and Thostenson [29] observed complex *TCR* dependences on temperature: two maxima with positive values, two minima with negative values for higher CNT fractions 0.5–1.0 wt.% and monotonic near-linear trend with positive slope and negative values for 0.1 wt.%. Negative *TCR* was observed by Karimov et al. [30] for MWCNT/glue nanocomposites. Similar results were observed by Nankali et al. [31] for MWCNT/PDMS silicone elastomer with non-linear *TCR* dependence. Negative *TCR*s with a complex dependence on temperature, having a maximum, were found by Xiao et al. [32] for a variety of filler fractions in MWCNT/epoxy nanocomposites.

Another complex *TCR* behaviour was reported by Gao et al. [33], where SWCNT/polyimide composite film was evaluated at a temperature range from room (25 °C) to annealing temperature up to 400 °C. As for fresh composite films, the initial *TCR* behaviour showed a negative trend. With increasing the annealing temperature, the *TCR* of films gradually changed from negative to positive, passing zero value.

Clearly, experimental results are controversial and demonstrate the play of several factors complicating the comparative analysis. As discussed above, many factors influence changes in the electrical resistivity of a CNT nanocomposite with temperature. Intrinsic conductivity of CNTs depends on temperature, thermal expansion of material causes changes in tunnelling distances while tunnelling thresholds depend on temperature as well. In addition to all these factors, we investigate the influence of another one–the degree of cure of the polymer matrix. 

Industrial practice tells us that several hours at elevated temperatures (3 h at 130 °C for the materials studied by us) provide a relatively high degree of cure. Does this degree of cure allow us to conduct *TCR* measurements? This is a well-posed question because, by conducting these measurements at elevated temperatures, we activate the post-curing process with the changes in electrical resistivity. These changes may shift the measured values of resistivity leading to biased *TCR*s. We argue that controversy in the literature on *TCR* values may be attributed to this effect.

Indirect proof of this statement comes from the results of *TCR*s measured for thermoplastics, where the post-curing does not take place. Mohiuddin and Hoa [34] observed a steady, near-linear increase of conductivity with temperature leading to monotonic *TCR* dependence with negative values for MWCNT/PEEK. Cen-Puc et al. [35] for MWCNT/PSF reported near linear resistance changes with temperature when slope (and thereby the sign of *TCR*) depended on the filler fraction–positive for <10 wt.% and negative for 50 wt.%. No extrema were observed. 

The present work aims to offer a better understanding of the influence of possible post-curing on the *TCR* values of the single-walled CNT/epoxy nanocomposite. For this purpose, curing durations longer than recommended by industrial practice are investigated. The manuscript addresses (1) the duration of the post-curing process marked by the saturation of changes in the electrical response of the material; (2) changes in *TCR* with the different post-curing stages; (3) the *TCR* of the fully post-cured material.

## 2. Materials and Methods

### 2.1. Materials

The single-walled CNT masterbatch used in the experiment was TUBALL™ MATRIX 201 (OCSiAl), specifically designed to provide superior electrical conductivity to epoxy, polyester, and polyurethane resins. To fabricate CNT/epoxy samples, the epoxy resin system Biresin CR131, designed for high-performance fibre-reinforced polymer composite applications, was used as the matrix. 

### 2.2. Samples’ Fabrication

CNT masterbatch was mixed with Biresin CR131 resin by shear mixing for desired 0.6 wt.% of CNTs at room temperature of 25 °C and relative humidity of 30%. Three different stirring cycles were utilized with varying speeds as presented in Table 1. The low vacuum of 0.1 mbar was applied for 15 min between each stirring cycle to reduce air entrapment. 

To ascertain the quality of nanofiller mixing, the scanning electron microscopy characterization of a sample fracture surface has been conducted (Figure 1). Apart from minor agglomeration and bundling present, the nanofiller demonstrates near-perfect distribution and dispersion, both isotropic and homogeneous.

To measure the electrical resistance of a sample, two copper tape electrodes were placed at the opposite sides of a cubic silicon mould 25 mm × 25 mm × 25 mm. Five samples for the constant temperature measurements and five samples for the varying temperature cycles were moulded and subsequently cured at an industrial temperature regime of 130 °C for 3 h (recommendation from resin manufacturer to achieve a high degree of cure).

### 2.3. Electrical Resistance Measurements at Post-Curing

DC electrical measurements were performed on CNT/epoxy samples using Keithley DMM6500 as shown in Figure 2. Before post-curing heating, the initial resistances of the samples were recorded. 

Procedure A: Five samples were post-cured at 120 °C in BINDER ED115 for 72 h uninterruptedly; electrical resistance measurements were conducted once a day with the samples maintained inside the oven. Every day, the duration of measurements was 1 h during which the change in readings was observed. Analysis of results is given in Section 3.1.

Procedure B: Five other samples were exposed to a temperature regime of 100 °C for 8 days, but with the heating interrupted for the *TCR* measurements in the range 25–100 °C. For this purpose, every cycle the samples were cooled down slowly to room temperature. Next, the electrical resistance of the samples was measured at temperatures 25 (room), 40, 60, 80, and 100 °C, consecutively, kept at the given temperature for 1 h at each step to equilibrate the temperature within the samples. After the end of the measurements, the samples were further post-cured at 100 °C for the remaining part of the day, and the next day the cycle repeated itself. This procedure was followed within 8 cycles. Analysis of results is given in Section 3.2.

## 3. Results and Discussion

In this section, we present the obtained results of the DC electrical resistance measurements conducted during the samples’ post-curing. 

### 3.1. Electrical Resistivity Change during Uninterrupted Post-Curing

The current discussion is for measurements conducted with Procedure A from Section 2.3. For the five samples, uninterruptedly post-cured at 120 °C for 72 h, the measured electrical resistivities are presented in Figure 3. Each connected set of markers represents the resistivity change of a sample during an hour. The decreasing trend is clearly observed, indicating the continuation of the curing process. We argue that the way the samples were manufactured (industrial curing at 130 °C for 3 h) does not allow them to reach a high enough degree of cure since post-curing clearly changes material properties. In Section 4, we compare the behaviour of post-cured samples in the presented work with reported investigations in the literature [13,29]. We see that only after 72 h at 120 °C, the curves reach a steady state indicating that the final state of curing has been achieved. 

All dependencies in Figure 3 possess identical trends but differ due to the scatter in the conductance values, remaining unchanged even after 72 h of post-curing. We attribute this scatter to variations in mixing, porosity, and variability of interface properties between nanocomposite mixture and copper electrode which can serve as an initiator of gas bubble nucleation as well as a modifier of CNT content due to the difference in CNT-polymer and CNT-copper surface energies. Nevertheless, the post-curing time dependencies present a clear trend, independent of the scatter present.

Table 2 represents the relative difference in electrical resistance of the samples. The table compares the initial resistance of the samples before heating with the resistances at the end of the thermal program. Due to Table 2, the average relative difference of the samples was obtained at 53% which implies a significant change. 

### 3.2. Changes in Electrical Resistivity with Temperature at Different Post-Curing States

The current discussion is for measurements conducted with Procedure B from Section 2.3. The dependence of the electrical resistance on temperature for the CNT/epoxy nanocomposite was investigated within 8 thermal cycles, each providing 24-h post-curing at 100 °C, interrupted for measurements. Experimental results are presented in Figure 4. Clearly, three trends can be observed: (1) the resistance decreases with post-curing until the material reaches the fully cured steady state (cycles 7 and 8); in cycles 1 and 2 the increase in resistance with temperature is not monotonic, exhibiting a peak, leading to the appearance of negative *TCR*s at higher temperatures; (3) in the fully cured state (cycles 7 and 8) the increase in resistance with temperature is monotonic, demonstrating only positive *TCR*s.

Utilizing these data to calculate *TCR*s, we depict results in Figure 5. Again, for the first cycles, when the degree of cure of the material is far from being fully cured, we observe the complex, sometimes monotonic, sometimes non-monotonic behaviour. However, in the fully cured state, the behaviour becomes stable, with the *TCR* monotonically increasing from a near zero value at room temperature to the values of the order of 0.001 at elevated temperatures.

Table 3 shows the relative difference in *TCR*s of the samples (the absolute value of the relative difference is not taken to demonstrate the crossing of zero). The table compares the *TCR*s at the end of cycles 1 and 8 for each sample. The results indicate the average relative difference of the samples is −394% which demonstrates that values at low degrees of cure are nearly unrelated to values at the fully cured state.

## 4. Comparative Analysis

To explore the influence of post-curing on resistivity and *TCR* of CNT nanocomposites, a comparison between the present study and experimental works in the literature was accomplished. Table 4 shows materials, temperatures, and CNT contents in compared studies (for comparison, we selected studies with a thermoset resin and full data for both resistivity and *TCR* temperature dependencies).

First, we compare temperature dependencies for resistivities. Both studies, [13,29], were conducted with MWCNTs. MWCNTs provide much higher resistivities than SWCNTs, by several orders of magnitudes higher [4], and, thereby, cannot be compared directly. To overcome this problem, we have normalized resistivities by their value at 25 °C to observe not absolute values but trends. In Figure 6a, we present these dependencies for sample 1 in our study and Figure 6b we put results from [13,29] on top of them. While Gong et al. [13] present a much wider scale of changes, beyond the limits of our plot, both with negative and positive slopes, Lasater and Thostenson data [29] follow our results closely–note the resemblance between Lasater and Thostenson data curves for 0.5 wt.%, 0.75 wt.%, 1.0 wt.% and cycle 1 curve (blue) from our results, corresponding to industrially cured materials (no post-curing). All these curves have a maximum of around 80 °C, which we attribute to active post-curing following the measurements and shifting the results.

In Figure 7, we compare *TCR* temperature dependencies. For our samples, *TCR*s exhibited maximums near 70 °C for initial cycles of post-curing, but were becoming steadily increasing monotonic dependencies at the end of post-curing. *TCR*s from [13] exhibited not only maxima but also minima near 50–80 °C, demonstrating much bigger absolute *TCR* values (Figure 7a). *TCR*s from [29] (Figure 7b) had values close to our study reaching zero values near 90 °C, similar to the cycle 1 dependence of the present study. 

We argue that the complex and non-monotonic behaviour of *TCR*s presented in the literature depends on the curing state of the material. Figure 5 demonstrates that in the fully cured state (cycles 7 and 8), as discussed in Section 3, the material shows stable, monotonically increasing dependence of *TCR* on temperature. 

## 5. Conclusions

The temperature coefficient of resistance (*TCR*) of a single-walled CNT/epoxy nanocomposite was studied considering the influence of post-curing. It was found that industrially cured material demonstrates post-curing resulting in complex, non-monotonic *TCR* behaviour. The reason is that during *TCR* measurements the samples are required to be equilibrated at the measurement temperature (1 h in the present study) which activates further post-curing and shifts the results. On the contrary, in the fully cured state, the material demonstrates stable, monotonically increasing dependence of TCR on temperature. We argue that the controversy of data on *TCR*s in the literature may be caused by the measurements taken not from fully cured material when measurements are influenced by the simultaneously initiated post-curing process. 

## Figures and Tables

**Figure 1 polymers-15-00433-f001:**
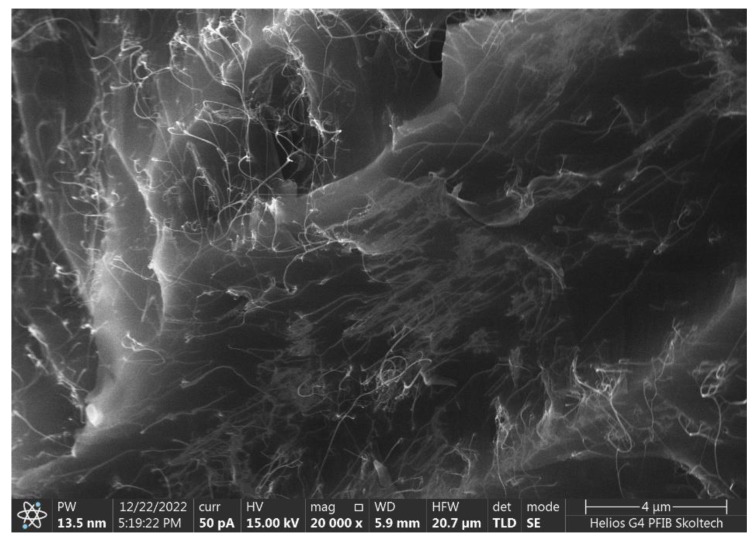
Scanning electron microscopy of a sample fracture surface.

**Figure 2 polymers-15-00433-f002:**
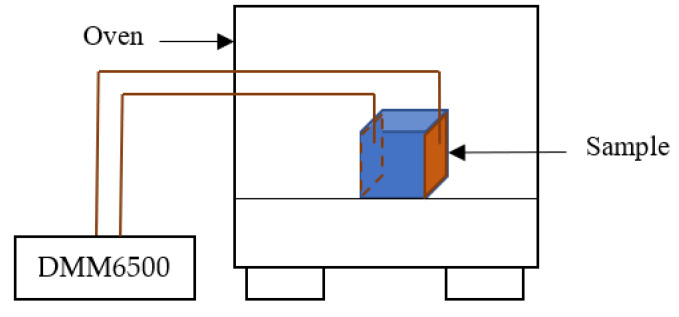
Schematic diagram of DC electrical measurements. Copper tape electrodes at the opposite faces of the sample to which conducting wires were soldered. The other end of conducting wires was connected to DMM6500 for DC electrical measurement.

**Figure 3 polymers-15-00433-f003:**
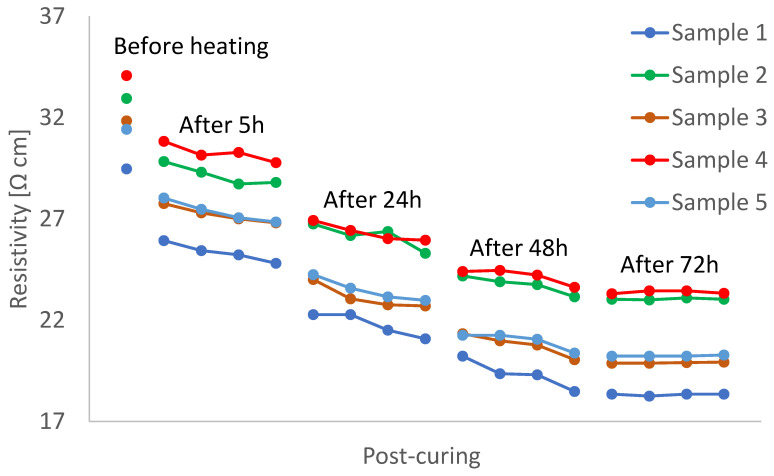
Change in electrical resistivity of single-walled CNT/epoxy samples during the post-curing at T = 120 °C for 72 h.

**Figure 4 polymers-15-00433-f004:**
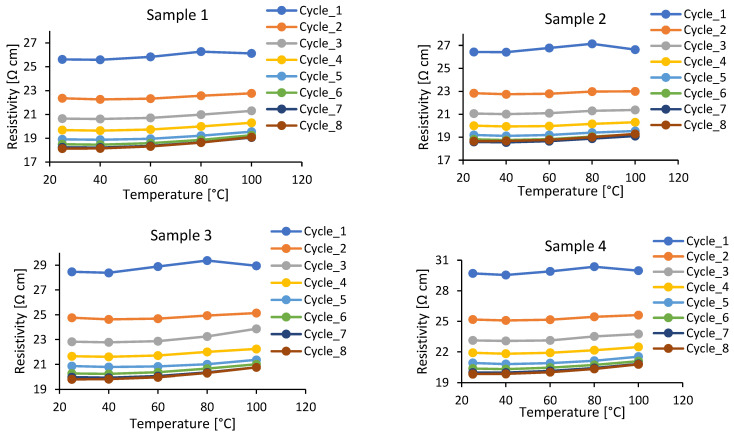
Change in electrical resistivity with the temperature at different post-curing cycles.

**Figure 5 polymers-15-00433-f005:**
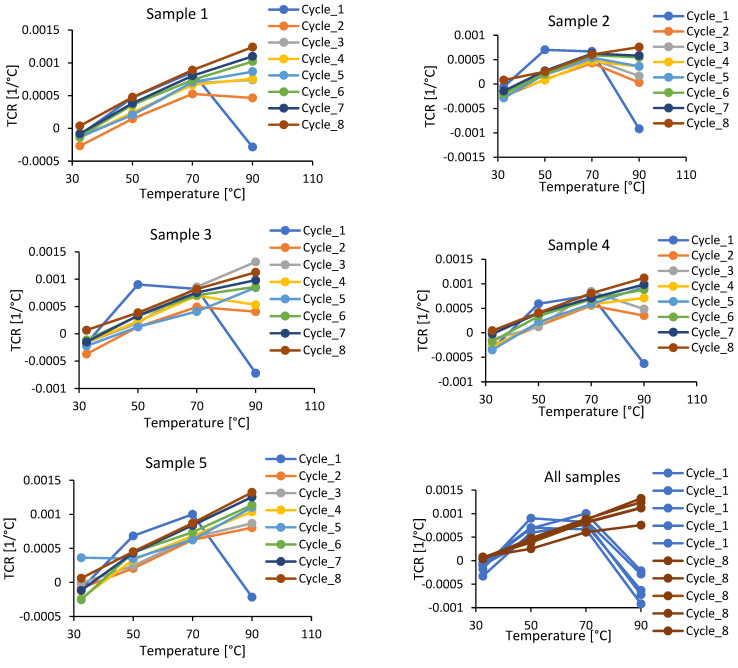
Change in *TCR* with the temperature at different post-curing cycles.

**Figure 6 polymers-15-00433-f006:**
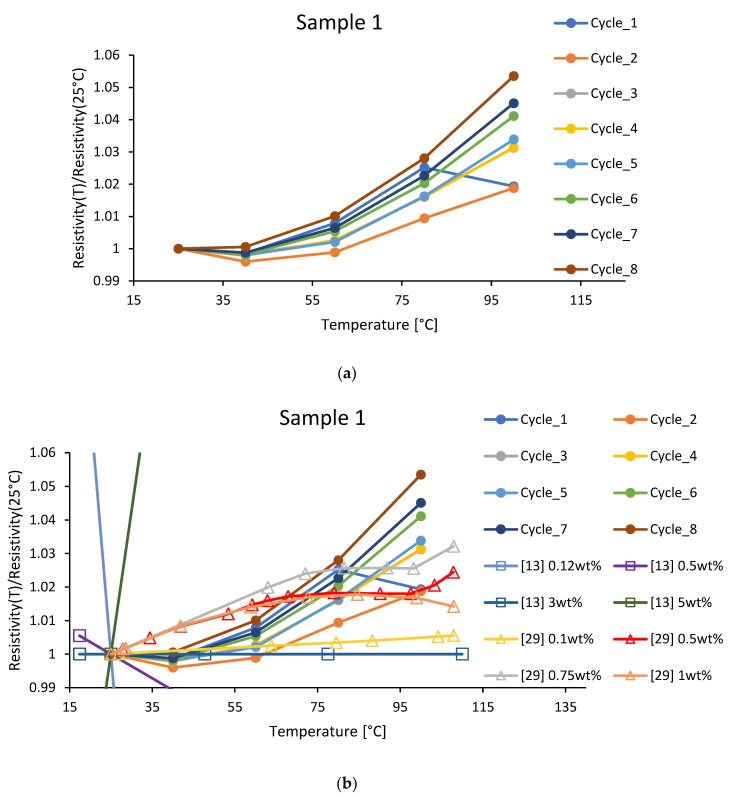
Relative resistivity temperature dependencies normalized by their values at 25 °C for (**a**) sample 1 in the current study (circles) with (**b**) dependencies from the literature added ([13]–squares, [29]–triangles).

**Figure 7 polymers-15-00433-f007:**
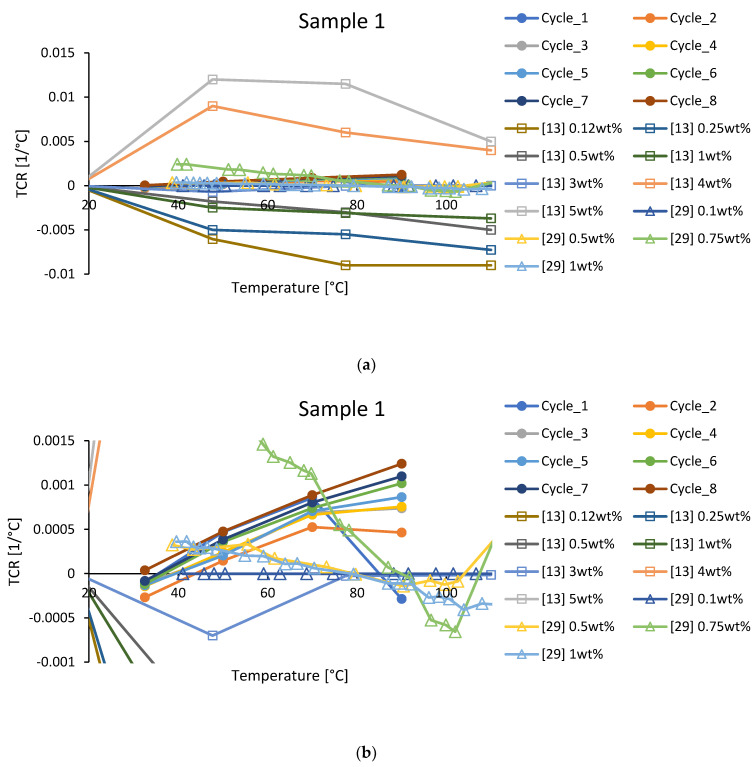
*TCR* comparison between sample 1 of the present study (circles) and literature data ([13]–squares, [29]–triangles), (**a**) full scale and (**b**) close-up.

**Table 1 polymers-15-00433-t001:** Steps of the synthesis process for CNT nanocomposite samples.

**Mixed Materials**	**Cycle 1**		**Cycle 2**		**Cycle 3**	
	Low speed + heating (45 °C) (20 min)	15 min Vacuum	High speed + heating (45 °C) (60 min)	15 min Vacuum	Low speed Without heating (20 min)	15 min Vacuum

**Table 2 polymers-15-00433-t002:** Relative difference in electrical resistance of epoxy-CNT single-walled with the influence of constant temperature.

Sample	Resistance before Heating [Ω]	Resistance at the End of Program [Ω]	Relative Difference
1	11.79	7.34	62%
2	13.18	9.21	43%
3	12.73	7.97	60%
4	13.63	9.33	46%
5	12.57	8.11	55%

**Table 3 polymers-15-00433-t003:** Relative difference in TCRs with the influence of curing cycles.

Sample	*TCR* at Cycle 1, 90 °C [1/°C]	*TCR* at Cycle 8, 90 °C [1/°C]	Relative Difference
1	−0.00028	0.0012	−535%
2	−0.00092	0.00076	−183%
3	−0.00072	0.0011	−257%
4	−0.00063	0.0011	−278%
5	−0.00021	0.0013	−717%

**Table 4 polymers-15-00433-t004:** Experimental material, temperature, and CNT content of the nanocomposites used in the literature and present study.

Source	Nanocomposite		Temperature [°C]	CNT Fraction
	CNT	Resin		
Present study	SWCNT	epoxy	25–100	0.6 wt.%
Gong et al. [13]	MWCNT	epoxy	−40–110	0.5 wt.%
Lasater et al. [29]	MWCNT	vinyl ester	25–170	0.5 wt.%

## Data Availability

The data presented in this study are available on request from the corresponding author.

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
