# Peer review of "Separating Curing and Temperature Effects on the Temperature Coefficient of Resistance for a Single-Walled Carbon Nanotube Nanocomposite"

_polymers, 2023, doi:10.3390/polym15020433_

Round 1
Reviewer 1 Report
To the authors:
The presented study by Jafarypouria et al. on the post-curing process does not provide structural/morphological/molecular and mechanical analysis of the post-curing treatment influence on the composite. Additional experiments are needed in order to add significance to the post-curing results on observed changes in TCR values.
Author Response
polymers-1930116 Jafarypouria M., Mahato B., Abaimov S.G
ANSWERS TO REVIEWERS
Reviewer 1 – For convenience, our revisions are highlighted in violet.
|
Reviewer’s comment |
Author’s answer |
|
The presented study by Jafarypouria et al. on the post-curing process does not provide structural/morphological/molecular and mechanical analysis of the post-curing treatment influence on the composite. Additional experiments are needed in order to add significance to the post-curing results on observed changes in TCR values. |
We thank the reviewer for pointing out that we are missing sample quality analysis and add Fig. 1 as SEM image demonstrating the quality of CNT dispersion. Simultaneously, we argue that other types of analysis are not relevant to our current research. We are interested in electrical conductivity changes as a diagnostic tool for post-curing, while mechanical changes from possible visco-elasticity to brittleness, changes in thermal conductivity, etc. represent other, possible future, studies, which are not a part of the current investigation. With regard to morphological and molecular studies, changes in electrical conductivity in CNT network are related to changes in tunnelling at distances less than 1nm which neither morphological SEM investigations, nor FTIR, nor other available tools can register because we are talking about changes within a 1nm gap or less in-between two CNTs. Unique such experiments are possible, but they present a separate long-term research exceeding the purpose of the reported results. |
Reviewer 2 Report
Comments are included in pdf

Author Response
polymers-1930116 Jafarypouria M., Mahato B., Abaimov S.G
ANSWERS TO REVIEWERS
Reviewer 2 – For convenience, our revisions are highlighted in gold.
|
Reviewer’s comment |
Author’s answer |
|
Comments are included in pdf |
We appreciate a positive evaluation of our results and have done the proposed modifications in the revised manuscript. |
|
functional surface groups (e.g., carboxylic, hydroxyl groups) |
Added. |
|
wt.-% |
Corrected. |
|
Strong difference in resistivity between different samples should be discussed (possible reasons, possible reduction of general validity). Even after 72h post-curing the differences remain on a high level (which shows this is not a curing state difference between the samples.
Nevertheless, for sure the measured trends during post-curing remain obvious and your main statement remains robust! |
Discussion added. Thank you for positive evaluation! |
|
2 significant decimal places? |
Corrected. |
|
Units |
Corrected. |
|
For all samples, both, resistivity and TEC, become much more stable after post-curing (cycles 7 + 8). Therefore, an additional comparison of TCR values after cycle 8 for all samples makes sense. |
Comparative figures are added into Figs.4-5 for cycle 1 and cycle 8. |
|
How is relative difference defined when zero-crossing is observed? |
Absolute value is not taken to demonstrate the crossing of zero. Clarifying statement is added. |
|
) |
Added. |
Reviewer 3 Report
The manuscript entitled "Separating curing and temperature effects on the temperature coefficient of resistance for a single-walled carbon nanotube nanocomposite" discusses the influence of post-curing on the TCR values of the single-walled CNT/epoxy nanocomposite. The manuscript needs a minor correction and cannot be accepted in its present form.
Below are some suggestions:
1. Figure 1 caption is on the next page. Please adjust the figure scale so that caption and figure can be displayed together.
2. In table 2, what is the unit of resistance?
3. Add literature in the result and discussion section to support your arguments.
4. Check the reference format.. use one out of two numbering system.
Author Response
polymers-1930116 Jafarypouria M., Mahato B., Abaimov S.G
ANSWERS TO REVIEWERS
Reviewer 3 – For convenience, our revisions are highlighted in green.
|
Reviewer’s comment |
Author’s answer |
|
The manuscript entitled "Separating curing and temperature effects on the temperature coefficient of resistance for a single-walled carbon nanotube nanocomposite" discusses the influence of post-curing on the TCR values of the single-walled CNT/epoxy nanocomposite. The manuscript needs a minor correction and cannot be accepted in its present form. Below are some suggestions: |
We appreciate a positive evaluation of our results and have done the proposed modifications in the revised manuscript. |
|
1. Figure 1 caption is on the next page. Please adjust the figure scale so that caption and figure can be displayed together. |
Corrected in our Word file. However, this depends on how the system generates system Word file from ours and will be corrected later at the proof stage. |
|
2. In table 2, what is the unit of resistance? |
Corrected. |
|
3. Add literature in the result and discussion section to support your arguments. |
References to studies with which we compare our results are added. To the best of our knowledge, we do not know literature sources suggesting that 72 hours of post-curing are required to do TCR measurements. And, typically, discussion with literature takes place in introduction, not in conclusions. If the Reviewer has particular refs relevant to the discussion, we will be happy to add them. |
|
4. Check the reference format.. use one out of two numbering system. |
At previous version, refs were done manually which was prone to errors. Now it is automated with EndNote. Also, we added DOI. |
Reviewer 4 Report
In this paper, authors investigated the influence of the degree of cure of polymer matrix on changing of electrical resistivity of a CNT nanocomposite with temperature. They showed dependence monotonically increasing from 25 to 100°C. The presentation of the problem, article structure and results are acceptable, but it needs revision for publication.
1- This sentence needs specific reference: “TCR of a CNT/polymer nanocomposite depends extremely on the morphology of CNT networks and properties of CNTs and polymer.”
2- As the authors mentioned the morphology of carbon nanotubes such as the dispersion and size have effects on the conductivity of CNT/epoxy but they did not mention the effect of these parameters in the samples fabricate in their study.
3- Humidity is an important factor in the electrical conductivity of CNTs. (Look at this article: DOI: 10.1109/JSEN.2020.3038647). It is necessary to discuss the effect of humidity on electrical resistance in the introduction or refer to the relevant reference in the manuscript.
4- Section 2.2. Samples’ fabrication: Have the materials been mixed at room temperature and room conditions? Mention the environmental conditions in the fabrication process.
5- Mention the exact value of the pressure in this sentence: “Low vacuum was applied for 15 min between each stirring cycle to reduce air entrapment.”
6- I recommend providing a schematic of the electrical resistance measurements setup.
7- No characterization method is provided for the fabricated samples. How are the authors sure of the correct preparation process of samples?
8- Section 3.2: It requires more detailed explanations of the differences between the cycles.
Additional comments:
- I recommend reducing the number of references [3-10] and [11-16] and keeping the ones that are more important.
- Remove the box around Figures 4 and 5.
Author Response
polymers-1930116 Jafarypouria M., Mahato B., Abaimov S.G
ANSWERS TO REVIEWERS
Reviewer 4 – For convenience, our revisions are highlighted in blue.
|
Reviewer’s comment |
Author’s answer |
|
In this paper, authors investigated the influence of the degree of cure of polymer matrix on changing of electrical resistivity of a CNT nanocomposite with temperature. They showed dependence monotonically increasing from 25 to 100°C. The presentation of the problem, article structure and results are acceptable, but it needs revision for publication. Below are some suggestions: |
We have done the proposed modifications in the revised manuscript. |
|
1- This sentence needs specific reference: “TCR of a CNT/polymer nanocomposite depends extremely on the morphology of CNT networks and properties of CNTs and polymer.” |
Added. |
|
2- As the authors mentioned the morphology of carbon nanotubes such as the dispersion and size have effects on the conductivity of CNT/epoxy but they did not mention the effect of these parameters in the samples fabricate in their study. |
Fig. 1 is added to demonstrate the dispersion of CNTs in the samples. |
|
3- Humidity is an important factor in the electrical conductivity of CNTs. (Look at this article: DOI: 10.1109/JSEN.2020.3038647). It is necessary to discuss the effect of humidity on electrical resistance in the introduction or refer to the relevant reference in the manuscript. |
The discussion added. |
|
4- Section 2.2. Samples’ fabrication: Have the materials been mixed at room temperature and room conditions? Mention the environmental conditions in the fabrication process. |
Added. |
|
5- Mention the exact value of the pressure in this sentence: “Low vacuum was applied for 15 min between each stirring cycle to reduce air entrapment.” |
Added. |
|
6- I recommend providing a schematic of the electrical resistance measurements setup. |
Fig. 2 is added. |
|
7- No characterization method is provided for the fabricated samples. How are the authors sure of the correct preparation process of samples? |
Fig. 1 is added, demonstrating CNT dispersion. Besides, the authors of study have long term experimental experience of mixing CNTs with epoxy (see, e.g. [4]). |
|
8- Section 3.2: It requires more detailed explanations of the differences between the cycles. |
This was in detail discussed in section 2.3. We added names Procedure A and Procedure B and cross-linked sections for better connection. |
|
- I recommend reducing the number of references [3-10] and [11-16] and keeping the ones that are more important. |
Done. |
|
- Remove the box around Figures 4 and 5. |
Done. |
Round 2
Reviewer 1 Report
The authors performed additional experiments and upgraded the quality of the manuscript. I disagree with the authors’ comments about molecular studies, since they are often employed in the field.
I support the publication of this study.